# MBP-11901 Inhibits Tumor Growth of Hepatocellular Carcinoma through Multitargeted Inhibition of Receptor Tyrosine Kinases

**DOI:** 10.3390/cancers14081994

**Published:** 2022-04-14

**Authors:** Hyun Jin Park, Garam Choi, Seongmin Ha, Yesl Kim, Min-Jin Choi, Minsup Kim, Md. Kamrul Islam, Yongmin Chang, Tae-Jun Kwon, Dongkyu Kim, Eunbee Jang, Tae Hwan Kim, Sha Joung Chang, Yeoun-Hee Kim

**Affiliations:** 1R&D Center, Etnova Therapeutics Corp., 124, Sagimakgol-ro, Jungwon-gu, Seongnam-si 13207, Korea; phj0808@etnova.co.kr (H.J.P.); garam1458@etnova.co.kr (G.C.); yesl_kim@etnova.co.kr (Y.K.); zzz9924@etnova.co.kr (M.-J.C.); sjchang3136@etnova.co.kr (S.J.C.); 2Institute of Biomedical Engineering Research, Kyungpook National University, 680, Gukchaebosang-ro, Jung-gu, Daegu 41944, Korea; 996574@naver.com (S.H.); mkislam2008@yahoo.com (M.K.I.); ychang@knu.ac.kr (Y.C.); 3InCerebro Drug Discovery Institute, Seoul 01811, Korea; minsupkim.bio@gmail.com; 4Preclinical Research Center, Daegu-Gyeongbuk Medical Innovation Foundation, 80, Cheombok-ro, Dong-gu, Daegu 41061, Korea; tjkwon@kmedihub.re.kr (T.-J.K.); dgkim728@kmedihub.re.kr (D.K.); 5College of Pharmacy, Daegu Catholic University, 13-13, Hayang-ro, Hayang-eup, Gyeongsan-si 38430, Korea; wkddmsql96@naver.com (E.J.); thkim@cu.ac.kr (T.H.K.)

**Keywords:** hepatocellular carcinoma, targeted therapy, tyrosine kinase inhibitor, anticancer agent, complete response

## Abstract

**Simple Summary:**

Although various treatments such as surgery and chemotherapy exist for advanced or unresectable HCC, most patients suffer from intractable diseases, having a poor prognosis. While immunotherapy using immune checkpoint inhibitors was recently proposed for HCC, only a small percentage of patients respond. Thus, there remains an unmet need for the development of therapeutic agents for the treatment of liver cancer. Here, we presented multi-RTKi MBP-11901, an innovative targeted anticancer agent for HCC, suggesting it as a new therapeutic strategy for the treatment of liver cancer.

**Abstract:**

Hepatocellular carcinomas (HCCs) are aggressive tumors with a poor prognosis. Approved first-line treatments include sorafenib, lenvatinib, and a combination of atezolizumab and bevacizumab; however, they do not cure HCC. We investigated MBP-11901 as a drug candidate for HCC. Cell proliferation and cytotoxicity were evaluated using normal and cancer human liver cell lines, while Western blotting and flow cytometry evaluated apoptosis. The anticancer effect of MBP-11901 was verified in vitro through migration, invasion, colony formation, and JC-1 MMP assays. In mouse models, the tumor volume, tumor weight, and bodyweight were measured, and cancer cell proliferation and apoptosis were analyzed. The toxicity of MBP-11901 was investigated through GOT/GPT and histological analyses in the liver and kidney. The signaling mechanism of MBP-11901 was investigated through kinase assays, phosphorylation analysis, and in silico docking simulations. Results. MBP-11901 was effective against various human HCC cell lines, leading to the disappearance of most tumors when administered orally in animal models. This effect was dose-dependent, with no differences in efficacy according to administration intervals. MBP-11901 induced anticancer effects by targeting the signaling mechanisms of FLT3, VEGFR2, c-KIT, and PDGFRβ. MBP-11901 is suggested as a novel therapeutic agent for the treatment of advanced or unresectable liver cancer.

## 1. Introduction

Hepatocellular carcinoma (HCC) is a dominant health problem. Detection of HCC at an early stage is difficult and most tumors are detected at an advanced stage due to their rapid progression and high metastasis, resulting in a poor prognosis for affected individuals. The main options for the clinical treatment of HCC include surgery, radiation therapy, transarterial chemoembolization, and molecular targeted therapy. First-line drugs include sorafenib (approved in 2007) and lenvatinib (approved in 2018), as well as the most recently approved combination therapy of atezolizumab and bevacizumab (approved in 2020). In addition, a number of second-line or tertiary drugs, which are targeted small molecular drugs, such as cabozantinib (2019), regorafenib (2017), the monoclonal antibody ramucirumab (2019), and the immune checkpoint inhibitors nivolumab (2017) and pembrolizumab (2018), have been approved by the U.S. Food and Drug Administration (FDA) as a follow-up treatment for patients pretreated with sorafenib [1]. However, there are limitations in the treatment of HCC. Most patients develop resistance to sorafenib within 6 months, resulting in a poor response rate. In the case of lenvatinib, follow-up treatment is impossible due to the absence of an approved second-line treatment. As immune checkpoint inhibitors have few side effects, their demand has surged worldwide over the past 5 years. However, even these have now been associated with a novel spectrum of side-effects, such as autoimmune-related adverse events [2]. Despite all these possible treatments, the median progression-free survival (PFS) in patients with HCC is less than 1 year. Existing treatments do not completely treat liver cancer, but only prolong survival. Although advances in medical technology have increased the promising 5-year survival rate of patients with HCC, long-term survival rates after surgical resection remain low due to high recurrence and metastasis rates [3,4]. Thus, there is an urgent need to develop new evidence-based therapeutics for the treatment of HCC.

In general, the pathogenesis of HCC includes the induction of tumor growth, metastasis, and angiogenesis through the binding of growth factors to cell surface receptors, which activate the proliferative signaling system. The development of therapeutics targeting such signal transduction pathways has been actively studied over the last years. For instance, a multiple kinase inhibitor (TKI), sorafenib (Nexavar^®^), was reported to prolong the survival period of patients with advanced HCC [5]. In addition, various targeted therapeutics have been studied, raising expectations for the treatment of refractory HCC. The advent of TKIs has become a promising targeted therapeutic strategy [6,7,8]. Briefly, TKIs enter cells and interact with several receptors and other intracellular signaling molecules to block the phosphorylation of tyrosine residues and hence the activation of various downstream signaling pathways, such as Ras/Raf/MEK/MAPK and PI3K/AKT/mTOR [9].

Recently, a new series of benzothiazole aniline derivatives have been synthesized in our laboratory, and among them, MBP-11901 (Figure 1A) showed excellent inhibitory activity against HCC cell growth in vitro [10]. The present study investigated the efficacy of this novel orally administered TKI, MBP-11901, against human HCC in vitro and in vivo with the aim of ultimately characterizing its mechanism of action and its therapeutic potential in treating HCC.

## 2. Materials and Methods

### 2.1. Chemicals

MBP-11901 (Figure 1) was prepared at 99.64% purity as previously reported [10] with further modifications for pilot-scale production. HPLC analysis was performed using a Prominence model (Shimadzu Corporation, Kyoto, Japan) equipped with a photodiode array detector (SPD-M20A) to confirm the purity of MBP-11901 (Appendix A). Test sample solution was prepared in water at a concentration of 200 ppm, and a SunFire C18 column (100 Å, 5 µm, 4.6 mm × 250 mm) was used for analysis. Mobile phases consisted of 0.1% formic acid-buffered water (A) and acetonitrile (B), with the gradient mode at a flow rate of 1 mL/min. Injection volume was 10 µL and the wavelength was fixed at 310 nm for UV detection. Data were acquired based on the following gradient system: T_min_/B%—T_0_/10, T_1.5_/10, T_4.5_/60, T_10.5_/80, T_15_/80, T_28.5_/10, T_30_/10.

### 2.2. Reagents

Eagle’s minimum essential medium (EMEM) was obtained from ATCC (Cat. 30-2003, Manassas, VA, USA). RPMI-1640 medium was obtained from Welgene (Cat. LM011-01, Daegu, Republic of Korea). Fetal bovine serum (FBS) was obtained from Hyclone (Cat. SH30919.03, Waltham, MA, USA). Antibiotic–antimycotic solution was obtained from Hyclone (Cat. SV30079.01). Antibodies against β-actin (Cat. sc-47778) were purchased from Santa Cruz Biotechnology (Dallas, TX, USA). Antibodies against phospho-FLT3 (Cat. #3461), Ras (Cat. #3965), phospho-c-Raf (Cat. #9421), phospho-MEK1/2 (Cat. #9121), phospho-ERK1/2 (Cat. #9101), phospho-Akt (Cat. #9271), phospho-mTOR (Cat. #5536), phospho-VEGFR2 (Cat. #3770), caspase-3 (Cat. #9662), cleaved caspase-3 (Cat. #9661), and PARP (Cat. #9542) were purchased from Cell Signaling Technology (Beverly, MA, USA). Phospho-PDGFRβ (Cat. LF-PA0035) was obtained from BioVendor Laboratory Medicine Inc. (Brno, Czech Republic).

### 2.3. Cell Lines and Culture

All cancer cell lines and corresponding normal cell lines were purchased from the American Type Culture Collection (ATCC). Hepatocellular carcinoma cell lines, HepG2 (HBV(−), p53 wt, ATCC^®^ HB-8065™) and Hep3B (HBV(+), p53 null, ATCC^®^ HB-8064™) were obtained from ATCC. Huh-7 (HBV(−), p53 mut., KCLB No. 60104) and PLC/PRF.5 (HBV(+), p53 mut., KCLB No. 28024) cells were obtained from the Korean Cell Line Bank (Seoul National University, Seoul, Korea).

Human colorectal adenocarcinoma cells (HT-29, ATCC^®^ HTB-38™), adenocarcinoma human alveolar basal epithelial cells (A549, ATCC^®^ CCL-185), and prostate cancer cells (PC-3, ATCC^®^ CRL-1435) were cultured in RPMI1640 medium. Human breast adenocarcinoma cells (MCF-7, ATCC^®^ HTB-22™ and MDA-MB-231, ATCC^®^ HTB-26™), rattus brain glioma cells (C6, ATCC^®^ CRL-2303™), and human renal carcinoma cells (Caki-2, ATCC^®^ HTB-47™) were cultured in Dulbecco’s modified Eagle’s medium (DMEM, Cat. LM001-05, WelGENE, Daegu, Korea). Mouse brain neural cells (NE-4C, ATCC^®^ CRL-2925™) and human cervix adenocarcinoma cells (HeLa, ATCC^®^ CCL-2™) were cultured in EMEM (ATCC^®^ 30-2003™), while human embryonic kidney cells (HEK-293, ATCC^®^ CRL-1573™) were cultured in minimum essential medium (MEM, Cat. LM007-09, WelGENE). All media contained 10% (*v/v*) FBS and antibiotic–antimycotic solution (1 mL per 100 mL of cell culture medium). Mouse liver normal cells (AML12, ATCC^®^ CRL-2254™) were cultured in Dulbecco’s modified Eagle medium/nutrient mixture F-12 (DMEM/F-12, Cat. LM002-04, WelGENE) containing 10% FBS (Hyclone), 1× ITS (10 μg/mL insulin, 5.5 μg/mL transferrin, and 6.7 ng/mL selenium; Cat. 41400-045, Gibco, Carlsbad, CA, USA), 40 ng/mL dexamethasone (Cat. D4902, Sigma-Aldrich, St. Louis, MO, USA), and 1% antibiotic–antimycotic solution (Hyclone). Human breast epithelial cells (MCF 10A, ATCC^®^ CRL-10317™) were cultured in DMEM/F-12 (WelGENE) supplemented with 5% horse serum (Cat. 16050-122, Gibco), 20 ng/mL epidermal growth factor (EGF; Peprotech Inc., Rocky Hill, NJ, USA), 500 μg/mL hydrocortisone (Sigma-Aldrich), 100 ng/mL cholera toxin (Sigma-Aldrich), 10 μg/mL insulin (Cat. I9278, Sigma-Aldrich), and 1% antibiotic–antimycotic solution (Hyclone). Human colon normal epithelial cells (FHC, ATCC^®^ CRL-1831™) were cultured in DMEM/F-12 (WelGENE) supplemented with 10% FBS (Hyclone), 10 ng/mL cholera toxin (Sigma-Aldrich), 20 ng/mL epidermal growth factor (EGF; Peprotech Inc., Rocky Hill, NJ, USA), 100 μg/mL hydrocortisone (Sigma-Aldrich), 1× ITS (10 μg/mL insulin, 5.5 μg/mL transferrin, and 6.7 ng/mL selenium; Gibco) and 1% antibiotic–antimycotic solution (Hyclone). HepG2 and Hep3B were cultured in EMEM medium, whereas Huh-7 and PLC.PRF.5 cells were cultured in RPMI-1640 medium. All media contained 10% (*v/v*) FBS and antibiotic–antimycotic solution (1 mL per 100 mL of cell culture medium). Cells were maintained at 37 °C in a humidified incubator under an atmosphere of 5% CO_2_.

### 2.4. Cell Proliferation Assay

Cell proliferation was detected using the Cell Counting Kit-8 (CCK-8, Cat. CK04-13; Dojindo Laboratories, Kumamoto, Japan) according to the manufacturer’s instruction. To evaluate cytotoxicity, cells were seeded in a 96-well plate (FHC, 1 × 10^4^ cells/well; Caki-2, PLC/PFR.5, and MCF 10A, 1 × 10^4^ cells/well; AML12, HepG2, Hep3B, and Huh-7 1.2 × 10^4^ cells/well; HeLa, HEK-293, A549, and PC-3, 1.5 × 10^4^ cells/well; HT-29, MCF-7, C6, and NE-4C, 2 × 10^4^ cells/well). After attachment and stabilization of cells for 24 h, the medium was switched with fresh containing various concentrations (0, 1, 5, 10, 25, 50, and 100 μM) of MBP-11901, and cells were incubated for 22 h. Next, CCK-8 solution was added to each well, and cells were incubated for another 2 h. Absorbance was then measured at 450 nm using a microplate reader (SpectraMax i3, Molecular Devices, San Jose, CA, USA). The IC_50_ and log IC_50_ values were calculated using GraphPad Prism (version 5.02; GraphPad PRISM software Inc. San Diego, CA, USA). All experiments were independently performed 3 times. The graph of log IC_50_ values represents average values.

### 2.5. Western Blot Analysis

HepG2 cells were seeded in a 60 mm^2^ culture dish at a density of 1.5 × 10^6^ cells/dish and incubated for 24 h. Cells were treated with MBP-11901 for 24 h. Cells were harvested and protein extraction and Western blotting methods were performed as previously described [11]. Membranes were incubated using the following diluted primary antibodies; PARP (1:1000 dilution), caspase-3 (1:1000 dilution), cleaved caspase-3 (1:1000 dilution), pFLT3 (1:500 dilution), Ras (1:1000 dilution), pc-Raf (1:1000 dilution), pMEK1/2 (1:1000 dilution), pERK1/2 (1:1000 dilution), pAKT (1:1000 dilution), pmTOR (1:1000 dilution), pVEGFR2 (1:1000 dilution), pPDGFR β (1:2000 dilution), and β-actin (1:1000 dilution) in Tris-HCl-based buffer containing 0.2% Tween-20 (T1027, Biosesang, Seongnam, Korea) (TBS-T; pH 7.5). Membranes were then incubated with horseradish-conjugated secondary antibodies (1:5000, Cell Signaling Technologies, Beverly, MA, USA) for 1 h at 25 °C. After washing with TBS-T, immunoreactive bands were visualized using the chemiluminescence western imaging system (Supernova-Q1800TM, Centronics, Daejeon, Korea). Band intensities were measured using ImageJ (Version 1.50i; US National Institutes of Health, Bethesda, MD, USA) and were normalized to β-actin.

### 2.6. Transwell Migration and Invasion Assay

For the migration assay, HepG2 cells were seeded into the upper Transwell chamber at a density of 2 × 10^4^ cells and incubated for 24 h. For the invasion assay, 100 μL of Matrigel: serum-free media (1:4) mixture was placed on the filter membrane in the upper chamber and incubated (37 °C, 4 h) to seed cells. Cells were exposed to MBP-11901 and sorafenib for 2 h. Then, the medium in the upper chamber was changed to serum-free medium, whereas the medium in the lower chamber was changed to a medium containing 10% FBS. The medium was replaced once every 2–3 d. Following 6 d incubation, the upper surfaces of the filters were scraped 5 times with cotton swabs to remove nonmigrated cells, whereas cells crossing the membrane into the lower chamber were stained with 1% (*w**/v*) crystal violet for 30 min. Crystal violet was removed and cells were washed with tap water. For the migration assay, after drying at 25 °C, 10% acetic acid solution was added to cells, and left to react in a shaker for 30 min. Absorbance was measured at 590 nm using a microplate reader (SpectraMax i3, Molecular Devices). For the invasion assay, the number of migrated cells was counted.

### 2.7. Flow Cytometry Analysis of Cell Death

Approximately 1.5 × 10^6^ HepG2 cells were suspended in 100 mL PBS, and then 200 mL 95% ethanol was added while vortexing. Cells were incubated at 4 °C for 1 h, washed with PBS, and resuspended in 250 mL 1.12% sodium citrate buffer (pH 8.4, C8532, Sigma-Aldrich) supplemented with 12.5 μg RNase. Incubation was continued at 37 °C for 30 min. Cellular DNA was then stained by applying 250 mL propidium iodide (50 μg/mL) for 30 min at 25 °C. Stained cells were analyzed on a FACScan flow cytometer for relative DNA content based on increased red fluorescence.

### 2.8. LDH Cytotoxicity Assay

The LDH assay was performed using the D-Plus™ LDH cell cytotoxicity assay kit (Cat. LDH-1000; Dongin Biotech, Seoul, Korea). Briefly, HepG2 cells were seeded at 1.2 × 10^4^ cells per well and AML12 cells at 2.0 × 10^4^ cells per well in 96-well plates. After 24 h of incubation, MBP-11901 at indicated concentrations was added to each well in a final volume of 100 μL. After 24 h, floating cells in the supernatants were removed by centrifugation at 600× *g* for 5 min. Control cells were lysed by adding 10 µL lysis buffer before centrifugation. Each supernatant (10 µL) was transferred to a new well in a 96-well plate. Finally, 100 µL LDH reaction mixture (1:50 ratio of WST substrate to LDH assay buffer) was added and samples were incubated at 25 °C for 30 min. Absorbance at 450 nm was measured using a microplate reader. Three independent experiments were performed in duplicate at different time points.

### 2.9. Colony Formation Assay

HepG2 cells were seeded in a 6-well plate at a density of 2 × 10^4^ cells/well and incubated for 24 h. Cells were treated with MBP-11901 and sorafenib for 2 h and media was changed with fresh one. Cells were incubated for 8 d and the medium was replaced once every 2–3 d. The medium was carefully removed, cells were washed with PBS, and fixed with 100% methanol for 30 min. After removing methanol, cells were stained with 0.5% (*w*/*v*) crystal violet for 30 min, and washed with tap water. The plate was dried at 25 °C and images were captured under a microscope (Nikon Eclipse Ts2, Tokyo, Japan). The stained area was automatically measured using ImageJ. This experiment was independently performed 3 times. The area of the colony was calculated as a percentage of the total area of the well.

### 2.10. JC-1 Mitochondrial Membrane Potential Assay

Alterations in MMP were investigated utilizing the cationic dye tetraethylbenzimidazolylcarbocyanine iodide (JC-1) according to the manufacturer’s instructions (Cat. ab113850, Abcam, Cambridge, MA, USA). HepG2 cells were seeded in a 35 mm^2^ dish at a density of 7.5 × 10^5^ cells/dish and incubated for 24 h. Cells were treated with MBP-11901 for 2 h and media was changed with fresh one. Briefly, after 2 d incubation, cells were washed with 1× dilution buffer, and incubated with 1 μM JC-1 in 1× dilution buffer for 30 min at 37 °C. Cells were then washed 3 times with 1× dilution buffer. Fluorescence was measured under a fluorescence microscope (ECLIPSE Ti, Nikon, Melville, NY, USA) using an excitation filter of 530 nm ± 15 nm and an emission filter of 590 nm ± 17.5 nm. The fluorescence intensity of immunostaining was quantified using ImageJ.

### 2.11. Immunofluorescence Staining

Sectioned tumor tissues (5 μm) were blocked with TBS supplemented with 5% normal goat serum (NGS) and 5% BSA for 2 h, and incubated overnight with primary antibodies (anti-caspase-3; diluted 1:100) in blocking solution (5% NGS and BSA) at 4 °C. Sections were then washed 3 times in TBS, and incubated for 1 h in the presence of Alexa Fluor-conjugated IgG-labeled secondary antibodies (Invitrogen, Carlsbad, CA, USA). Sections were washed and mounted with Vectashield mounting medium containing DAPI (Vectashield H-1500; Vector Laboratories, Inc., Burlingame, CA, USA). Images were captured using a fluorescence microscope (ECLIPSE Ti, Nikon) and fluorescence intensity of immunostaining was quantified using ImageJ.

### 2.12. Immunohistochemistry Staining

Sectioned tumor tissues (5 μm) were deparaffinized in xylene (Junsei Chemical, Tokyo, Japan) and rehydrated through an ethanol series. Endogenous peroxidase activity was inactivated by incubation in 0.3% H2O2 (Sigma-Aldrich) in methanol for 10 min. Sections were then rinsed in 0.1 M TBS (pH 7.4) and boiled in citrate buffer (pH 6.0) containing 0.03% Tween-20 for 4 min. Finally, sections were incubated with blocking solution (5% NGS and BSA in TBS) at 25 °C for 1 h, and subjected to indirect immunohistochemistry using an antibody against anti-Ki-67(diluted 1:100) for 1 h. For the negative control, the primary antibody was omitted and slides were incubated with a blocking solution. Sections were then incubated with HRP-conjugated anti-rabbit IgG for 1 h at 25 °C, stained with VECTOR1 NovaRED (Vector Laboratories, Inc.), and counterstained with hematoxylin (BBC Biochemical, Mount Vernon, WA, USA). Sections were dehydrated, cleared, and mounted with Permount (Fisher, Fair Lawn, NJ, USA). Images were captured under a fluorescence microscope (ECLIPSE Ti, Nikon) and the fluorescence intensity of immunostaining was quantified using ImageJ.

### 2.13. Animals

Male Balb/c-nude mouse weighing 20–22 g (Nara Biotech, Seoul, Republic of Korea) were used for all experiments. Animals were housed under controlled environmental conditions in an ambient temperature of 23 ± 1 °C and relative humidity of 50 ± 10% under a 12 h light/dark cycle and were given ad libitum access to food and water. All animal experiments were approved by and performed in accordance with the guidelines issued by the Institutional Animal Care and Use Committee (IACUC) of Kyungpook National University (No. KNU-2018-0086; Daegu, Korea).

### 2.14. In Vivo Tumor Growth Assay Using Xenograft Mouse Model

For the generation of a tumor xenograft, HepG2 cells (5 × 10^6^) were suspended in 100 μL serum free EMEM:Matrigel (1:1) and then subcutaneously injected into the upper-left flank region of nude mice. When the tumor reached a mean size of about 100–150 mm^3^, mice were randomized into each experimental group according to tumor size. Mice were orally administered Nexavar^®^ (82, or 164 mg/kg, daily or once every 2 d, respectively), MBP-11901 (40, 60, or 82 mg/kg, once every 2, 3, or 4 d, respectively), or saline as control (once every 2 d). Tumor size was measured with a caliper rule twice a week and body weight was measured simultaneously. The tumor size was calculated as follows: tumor size (mm^3^) = (L × W^2^)/2, where L is the longest and W is the shortest radius of the tumor in millimeters. At the end of the experiment, mice were euthanized, blood samples were collected via the abdominal vena cava, and tumors were removed and weighed.

Tumor, liver, and kidney tissues were harvested for fixation in 4% paraformaldehyde (PFA, BP031, Bio-solution, Suwon-si, Gyeonggi-do, Korea) for histologic examination and immunohistochemical staining or immediately frozen at −80 °C in a deep-freezer and stored until protein extraction.

Drug efficacy was expressed as the percentage of tumor growth inhibition (% TGI), calculated using the equation: 100 − (T/C × 100), where T is the mean relative tumor volume (RTV) of the treated tumor and C is the mean RTV in the vehicle control group at the time of sacrifice. RTV = V_x_/V_1_, where V_x_ is the volume in mm^3^ at a given time and V_1_ at the start of treatment. Mean TGI (%) and standard deviation were calculated for each group.

### 2.15. Measurement of Glutamic-Oxaloacetic Transaminase (GOT) and Glutamic-Pyruvic Transaminase (GPT)

The levels of GOT and GPT were measured using enzymatic assay kits (Asan Pharm, Seoul, Korea). The collected blood was allowed to clot and the serum, which was separated by centrifugation at 4000 rpm for 15 min after 2 h at 25 °C, was used for analysis.

### 2.16. In Vivo Tumor Growth Assay Using an Orthotopic Mouse Model

The orthotopic HCC mouse model was established by placing a small piece of tumor-derived HepG2-Luc cells. Mice were randomly distributed into 3 groups upon bioluminescence signal reaching 10^7^. The first group served as the saline control. The second and third groups were orally administered MBP-11901 (82 mg/kg) and Nexavar^®^ (164 mg/kg) once every other day for 4 weeks. Tumor progression/regression was recorded once weekly by quantifying the bioluminescence signals. Experimental animals were sacrificed by CO_2_ inhalation.

### 2.17. Histological Analysis

Tumors were fixed in 4% PFA for 72 h, and treated with an alcohol concentration gradient (50, 70, 95, and 100%), xylene (Junsei Chemical), and paraffin for 30 min, respectively. Sectioned tumor tissues (5 μm) were treated with xylene and an alcohol concentration gradient (100, 95, 70, and 50%) for 1 h and 10 min, respectively, in an oven at 65 °C. Hematoxylin and eosin (H&E) staining (BBC Biochemical, Mount Vernon, WA, USA) was performed according to the manufacturer’s instructions and stained tissues were observed under a microscope (ECLIPSE Ti, Nikon). For Sirius Red staining, tissues were immersed in Picro Sirius Red Stain (ab150681; Abcam) for 30 min, washed for 1 min in 0.1 N HCl, followed by dehydration and mounting of slides.

### 2.18. In Silico Molecular Docking Analysis

Crystal structures of FLT3, VEGFR2, MEK5, and KIT were obtained from the RCSB Protein Data Bank website (https://www.rcsb.org) (accessed on 21 July 2021) under the PDB IDs of 4RT7, 3VHE, 3SLS, and 4U0I, respectively. Due to the lack of an experimentally revealed structure, the three-dimensional structure of PDGFRβ was obtained from the AlphaFold Protein structure database (https://alphafold.ebi.ac.uk/entry/P09619) (accessed on 21 July 2021). These structures were optimized for in silico simulation using the Prime protein modeling application [12]. Protein–ligand docking simulation was performed using the Glide docking application [13]. Glide searches all possible binding poses of a given ligand and calculates the binding energy of the ligand using the GlideScore scoring function. For docking simulation, the flexible ligand sampling method was applied, using the SP (Standard Precision) mode of Glide.

### 2.19. KINOMEscan Profiling of MBP-11901

The profiling of MBP-11901 against a panel of 468 kinases was performed by DiscoverX using KINOMEscan technology, an active-site-dependent competition-binding assay (Eurofins DiscoverX Corporation, San Diego, CA, USA). Binding constants (Kd) were calculated by DiscoverX using KdELECT technology, a widely used assay for comparing selectivity across the kinome. The DiscoverX platform reports the thermodynamic Kd to facilitate direct comparison of inhibitor affinity across the kinome, independent of ATP concentration.

### 2.20. Statistical Analysis

Data were evaluated using one-way or two-way ANOVA followed by Tukey’s or Dunnett’s test. Analyses were performed using GraphPad Prism (version 5.02; GraphPad Prism Software Inc.). Data are expressed as the mean ± SD (standard deviation) or standard error of the mean (SEM), and *p* < 0.05 was considered statistically significant.

## 3. Results

### 3.1. MBP-11901 Showed the Most Potent Cytotoxicity to Human HCC Cells

We conducted a cell-based screening, examining the effects of MBP-11901 on the viability of various tumor and normal cells (Figure 1B and Appendix A). We used various concentrations of MBP-11901 ranging from 1 to 100 μM to treat cells for 24 h and determined cell viability using the CCK-8 assay. We found that MBP-11901 resulted to significant cytotoxicity in hepatocellular carcinoma and colon adenocarcinoma cells compared with corresponding normal cells, with a more dominant effect observed in hepatocellular carcinoma cells. We additionally confirmed the anticancer effect of MBP-11901 in various liver cancer cell lines, including HepG2, Hep3B, Huh-7, and PLC/PRF5 (Table 1). The half-maximal inhibitory concentration (IC_50_) values of MBP-11901 in HepG2, Hep3B, Huh-7, and PLC/PRF5 were 5.16 ± 1.37, 16.82 ± 0.88, 29.41 ±1.13, and 18.55 ±0.81 μM, respectively. As HepG2 cells exhibited the lowest IC_50_ values and were the most sensitive to MBP-11901 among the four cancer cell lines tested, we used HepG2 cells in our subsequent experiments.

### 3.2. MBP-11901 Induced Anticancer Effect on HepG2 Human HCC Cells

To test whether treatment with MBP-11901 induces the apoptosis of hepatocellular carcinoma cells, we evaluated the levels of protein expression of apoptosis markers in HepG2 cells, such as cleaved PARP1 and active caspase-3. Our results showed that treatment with MBP-11901 led to high levels of cleaved PARP and active caspase-3 in a time- and dose-dependent manner (Figure 2A,B), suggesting that MBP-11901 induced the apoptosis of HepG2 (Appendix A). Both migration and invasion are basic characteristics of malignant tumor cells. We thus performed Transwell assays to determine the effect of MBP-11901 on the migration and invasion ability of tumor cells. As shown in Figure 2C, MBP-11901 highly inhibited the migration and invasion of tumor cells in a dose-dependent manner. We also analyzed the cell cycle to further determine apoptotic characteristics. We accordingly detected that MBP-11901 caused an increase in the sub-G1 phase, indicating apoptosis in a time-dependent manner (Figure 2D).

### 3.3. MBP-11901 Was More Effective in Liver Cancer Cell Lines and More Toxic in Normal Liver Cell Lines Compared with Sorafenib

To compare the effectiveness of MBP-11901 and sorafenib, a major component of the primary treatment regime of clinical liver cancer, we performed cell viability assays using various liver cancer cell lines (HepG2, Hep3B, Huh-7, and PLC/PRF.5) and a normal liver cell line, AML12. We observed that MBP-11901 showed a similar level of anticancer effect in liver cancer cell lines as that shown by sorafenib (Figure 3A and Appendix A). However, in normal liver cells, sorafenib showed cytotoxicity even at low concentrations (IC_50_ = 13.44 ± 2.77 μM), whereas the cytotoxicity of MBP-11901 was significantly reduced (IC_50_ = 65.70 ± 1.62 μM), as indicated by the LDH assay (Figure 3B,C). In addition, western blot analysis confirmed a significant difference in the induction of apoptosis between MBP-11901 and sorafenib. We further confirmed that MBP-11901 increased the activity of PARP and caspase-3 compared with sorafenib (Figure 3D–F). The colony formation assay is a widely used method to study the number and size of cancer cell colonies that remain after irradiation or administration of cytotoxic agents and serves as a measure of the antiproliferative effect of these treatments [14]. We found that treatment with MBP-11901 resulted in a significant reduction in colony formation even at a concentration of 5 μM. In particular, we found that, compared with sorafenib, at high concentrations of MBP-11901 almost no colony growth was observed (Figure 3F).

We also measured the mitochondrial membrane potential of cells exposed to MBP-11901 using JC-1 staining. As shown in Figure 3G, the nontreated control group exhibited red fluorescence after administration of the JC-1 probe, indicating a high mitochondrial potential. In comparison, MBP-11901-treated cells exhibited high green fluorescence, indicating a decrease in mitochondrial potential.

### 3.4. Oral Administration of MBP-11901 Exhibited Excellent Anticancer Effects in HepG2 Cell-Derived Subcutaneous Xenograft Tumors in Nude Mice

To confirm the antitumor effect of MBP-11901 in vivo, we generated a HepG2 xenograft tumor model in BALB/c nude mice. A schematic timeline of this experiment over 6 weeks is shown in Figure 4A. We found that tumor growth was significantly inhibited in the MBP-11901 group (Figure 4B). The results of this experiment are summarized in Table 2. All animals survived throughout the experiments. We noticed that administration of 40, 60, and 82 mg/kg MBP-11901 for 19 d caused 61.41 ± 0.45, 85.26 ± 0.36, and 92.54 ± 0.02% inhibition of tumor growth, respectively, as compared with vehicle control. In contrast, the Nexavar^®^ (164 mg/kg) group showed a tumor growth inhibition of 38.77 ± 0.37% (Figure 4C). Similarly, we observed that tumor weights per mouse in the 40, 60, and 82 mg/kg MBP-11901-treated groups were 0.082 ± 0.063, 0.025 ± 0.035, and 0.011 ± 0.009 g, respectively, compared with 0.31 ± 0.089 g in the vehicle control group. In comparison, the tumor weight in the Nexavar^®^ group (164 mg/kg) was 0.144 ± 0.073 g (Figure 4E). We also found that the average body weight was decreased by 7.21 +3.41% in the Nexavar^®^ group, whereas only a weak weight loss was observed in the MBP-11901 groups (Figure 4D and Table 2). Analysis of the expression of Ki-67 and cleaved caspase-3 in tumors treated with MBP-11901 showed a significant reduction in the proliferation of tumor cells, and a high level of apoptosis compared with the vehicle control (Figure 4F,G).

### 3.5. Administration of MBP-11901 Led to the Complete Disappearance of the Tumor

We then confirmed the anticancer effect of MBP-11901 by designing a different administration interval. We orally administered MBP-11901 at a concentration of 82 mg/kg once every 2, 3, and 4 d, whereas 164 mg/kg Nexavar^®^ was administered every day or once every 2 d. The sacrifice date was matched, and drugs were administered for a total of 42 d. The total number of administrations according to each administration interval is summarized in Table 3. Images of either very small traces of tumors remaining or of tumors having completely disappeared in the subcutaneous tissue of mice treated with MBP-11901 regardless of the administration interval, are shown in Figure 5A. In the case of administration of Nexavar^®^, one mouse that was treated once every 2 d died (Figure 5A and Table 4). We confirmed that the antitumor effect was better when Nexavar^®^ was administered daily than when administered once every 2 d. Although we observed a significant difference in weight loss between subjects in the Nexavar^®^ group, no change in body weight was shown in the MBP-11901 group (Figure 5B). These changes in body weight (excluding tumor weight) were more obvious at 42 d (Appendix A). In particular, we found that treatment with 82 mg/kg MBP-11901 at 1T/2D, 1T/3D, and 1T/4D for 42 d caused 99.82 ± 0.33, 99.51 ± 0.81, and 99.48 ± 0.60% inhibition of tumor growth, respectively, compared with the saline-treated control (Figure 5C and Appendix A and Table 4), whereas, treatment with 164 mg/kg Nexavar^®^ at 1T/1D and 1T/2D for 42 d caused 85.68 ± 13.60 and 88.02 ± 4.53% inhibition of tumor growth, respectively, compared with the saline-treated control (Figure 5C and Appendix A and Table 4). Specifically, in the case of oral administration of MBP-11901, we noticed that the time required for the tumor to disappear was similar regardless of the administration interval (Figure 5C). We obtained our tumor weight results in a similar manner (Figure 5D). Despite individual differences in the serum levels of the GOT and GPT liver-specific enzymes, we detected that both the MBP-11901 1T/3D and 1T/4D and Nexavar^®^ groups were characterized by GPT values that were significantly outside the normal range (Figure 5E). However, histological analysis of liver and kidney tissues did not reveal any specific findings (Figure 5F).

### 3.6. Oral Administration of MBP-11901 Exhibited Excellent Anticancer Effects against Liver Tumors in the HepG2 Tumor-Bearing Orthotopic Animal Model

To verify the anticancer effect of MBP-11901 on tumors generated in liver tissues in vivo, we used an orthotopic HCC mouse model (Figure 6). We applied the same protocol as the one used in the subcutaneous transplantation model (once every 2 d, orally), and measured luminescence and body weight once a week. We performed a total of 14 administrations for a total of 28 d. We observed that at the first and second week of administration, the bioluminescence intensity of physiological saline (4.55 ± 10.1, 10.80 ± 20.7-fold, respectively) and Nexavar^®^ (4.73 ± 6.9, 9.05 ± 11.2-fold, respectively) was steadily increased without significant differences between the two groups (Figure 6A,B and Appendix A). We found that the bioluminescence intensity of MBP-11901 was slightly increased by 2.24 ± 2.9-fold in the first week, but decreased significantly to 0.83 ± 1.8 fold in the second week. At 4 weeks, the intensity in the saline group was increased by 26.92 ± 30.3-fold and in the Nexavar^®^ group by 11.69 ± 10.0 fold, whereas in the MBP-11901 group it was decreased by 0.02 ± 0.0-fold, so that almost no tumor bioluminescence was measured (Figure 6A,B and Appendix A). We did not detect any significant differences in the body weight between groups (Figure 6C). Moreover, although we noticed individual differences in the Nexavar^®^ group, a significant increase was detected in the level of GPT. In contrast, the MBP-11901 group showed values within the normal range (Figure 6D,E).

### 3.7. MBP-11901 Induced Anticancer Effects on HCC through Inhibition of Multitarget Tyrosine Kinase, FLT3, VEGFR2, PDGFRβ, and c-KIT

To identify the kinase targeted by MBP-11901 and determine its binding constant, we tested a single concentration of 10 μM against all 468 kinases, including mutants. As shown in Figure 7A, our kinase results for the primary screen binding interaction with MBP-11901 were reported as percent control (%Ctrl). In particular, %Ctrl indicated that the dissociative kinases (unbound to MBP-11901) were the percentages of all tested kinases, where lower numbers indicated stronger binding to MBP-11901. We hence identified six selective and activated kinase targeting of FLT3 (including all mutant forms), MEK5, MAP4K2, KIT, VEGFR2, and PDGFRβ within 35% of the assay results. Subsequently, we measured the binding interactions with MBP-11901 using KdELECT (KINOMEscan™, DiscoveRx, USA). As shown in Figure 7B, MBP-11901 bound selectively and tightly to FLT3, MAP4K2, KIT, VEGFR2, and PDGFRβ with Kd values of 295 ± 5, 1100 ± 0, 3850 ± 1150, 3550 ± 250, and 4000 ± 220 nM, respectively. The Kd values of MEK5, which showed high affinity in %Ctrl, and EGFR, FGFR1, and FGFR2, which were further analyzed (highly correlated with cancer proliferation and metastasis), were not significant.

We further investigated the potential mechanism of HCC apoptosis by MBP-11901 (Figure 7C). To this end, we evaluated the level of phosphorylation of Tyr591 of FLT3, Tyr1175 of VEGFR2, Tyr703 of c-KIT, Y716 of PDGFRβ, Ser259 of c-Raf, Ser217/221 of MEK1/2, Thr202/Tyr204 of ERK1/2, Ser473 of AKT1/2, and Ser2448 of mTOR using Western blotting. In particular, we confirmed that kinase phosphorylation was inhibited in HepG2 cells after exposure to MBP-11901 from 10 to 120 min (Figure 7C) in a concentration-dependent manner (Figure 7D). We also confirmed the inhibition of the phosphorylation of FLT3, VEGFR2, c-KIT, and PDGFRβ and that of their downstream signals MEK1/2, ERK1/2, and mTOR, in HepG2-subcutaneous xenograft mice orally administered MBP-11901 (Figure 7E).

We further verified the binding affinity of MBP-11901 to target kinases through in silico studies (Figure 7F). As in the KdELECT analysis, we found that MBP-11901 showed high binding affinity for FLT3, VEGFR2, and KIT with docking energies (ΔG) of −10.01, −8.926, and −8.335 Kcal/mol, respectively, whereas MBP-11901 exhibited a low binding affinity for MEK5 (−6.278 kcal/mol). Specifically, although we obtained clear results for KdELECT and phosphorylation inhibition, PDGFRβ showed low binding affinity (−6.857 kcal/mol).

Overall, from the results of our in vitro kinase assays, phosphorylation expression, and in silico binding affinity, we confirmed that MBP-11901 targeted FLT3, VEGFR2, c-KIT, and PDGFRβ in HCC (Figure 7G). We suggested that MBP-11901 is a targeted inhibitor of multiple tyrosine kinases, which blocks downstream signals by inhibiting their phosphorylation.

## 4. Discussion

Advanced or unresectable HCC is a malignant tumor with poor prognosis and few available treatment options [15]. Multikinase inhibitors such as sorafenib, lenvatinib, cabozantinib, and regorafenib have been suggested as first- and second-line therapy; however, the response to these drugs is low and limited. Recently, the number of options for the treatment of HCC has increased, with an added combination therapy with atezolizumab and bevacizumab [16]. Moreover, rather than reaching a full cure, these treatments merely prolong the overall survival period of patients by several months [16,17]. Surprisingly, immunotherapy of hepatocellular carcinoma has been reported to lead to the development of hyperprogressive cancer [18]. Therefore, it is necessary to carefully select patients with HCC for immunotherapy. In addition, there is still an unmet need for innovative therapeutic agents for the cure of HCC.

Here, we presented the excellent therapeutic effect of MBP-11901 on HCC using an animal model. We confirmed the HCC-specific efficacy of MBP-11901 through in vitro screening using various human cancer cell lines (Figure 1B), and verified its effectiveness in various HCC cell lines (Table 1). Through various experimental techniques, we verified that MBP-11901 not only induced the apoptosis of HCC cells, but also inhibited their proliferation, metastasis, and invasion (Figure 2 and Figure 3).

A noteworthy aspect of our study was the complete therapeutic effect of MBP-11901 in subcutaneous and orthotopic transplantation models. At the start of drug administration, the average tumor volume in each group was 200 mm^3^, as shown in Figure 4. was After oral administration once every 2 d, the average tumor volume in the Saline group reached approximately 1000 mm^3^ at the end of a total of 9 administrations, approximately 600 mm^3^ in the Nexavar^®^ group, whereas it reached approximately 350, 150, and 70 mm^3^, in the 40, 60, and 82 mg/kg MBP-11901 groups, respectively. In particular, in the 82 mg/kg MBP-11901 group, the tumor completely disappeared in 4 out of 8 animals; the same effect was observed in 3 out of 8 animals in the 60 mg/kg group. Histological analysis of MBP-11901-treated tumor masses revealed a decrease in the levels of proliferation proteins and an increase in those of apoptosis proteins.

Another noteworthy result shown in Figure 5 was that there was no difference in efficacy even if the dosing interval was changed. In this experiment, administration was initiated at an average tumor volume of approximately 190 mm^3^. Following termination of treatment after 42 d, tumors had disappeared in 3–4 out of a total of 6 animals in all groups to which MBP-11901 was applied. In the case of Nexavar^®^, the tumor suppression rate was approximately 2 times greater in the daily administration group than in the group treated once every 2 d. However, this effect was clearly lower than that observed in the group administered with MBP-11901 once every 4 d. When the interval between the administration of MBP-11901 was once every 3 d and once every 4 d, a statistically significant increase was found in the levels of GPT compared with normal values (Figure 5E); however, the weight loss rate was maintained at a similar level to normal (Table 4). Moreover, histological analysis of the liver and kidney showed no apparent toxicity (Figure 5F,G).

We further verified the effectiveness of MBP-11901 by constructing a mouse model in which Hep3B and Huh-7 were subcutaneously transplanted in addition to the human liver cancer cell line HepG2. In the case of Hep3B, the administration was initiated at a tumor volume of about 300 mm^3^, once every 2 d, and the experiment was terminated after a total of 12 administrations (Appendix A). The average tumor volume was measured to be approximately 1400 mm^3^ in the Saline group, whereas the Nexavar^®^ group had a tumor volume of about 900 mm^3^. In contrast, tumors had completely disappeared in 1 out of 6 animals in the MBP-11901 group; 1 had only a trace of a tumor measuring approximately 30 mm^3^, whereas the average tumor volume in the 6 animals was approximately 100 mm^3^. Compared with HepG2 or Hep3B, administration of MBP-11901 to the Huh-7 subcutaneously implanted model did not result in tumor disappearance, although the tumor weight was significantly reduced (Appendix A). These results showed that MBP-11901 did not show specificity for human HCC cell lines, but rather was effective against all evaluated HCC cell lines. Unlike Nexavar^®^, there were no side-effects relating to drug resistance or rapid tumor growth.

In addition to the HCC subcutaneous transplantation model, administration of MBP-11901 led to similar results of tumor elimination in the orthotopic mouse model as well (Figure 6). In order to determine the presence of a tumor, we obtained liver tissues at the end of the experiment and performed histological analysis of the orthotopic graft sites (Appendix A). When saline was administered, the boundary between the mouse liver and transplanted liver was clearly observed, as well as a portion positive for the antibody against Ki-67, a proliferation marker. In contrast, when MBP-11901 was administered, the boundary between the mouse liver and transplanted tissue was indistinguishable, and almost no cells were observed to be stained with Ki-67 and hematoxylin. Sirius Red staining confirmed that the traces still remaining were collagen fibers similar to scars. In the case of the HepG2 subcutaneous implant, when the skin in the area where the tumor was present was incised and observed with the naked eye at autopsy, no tissue mass was detected and only traces remained similar to the wound (Appendix A). These findings suggested the complete disappearance of HCC in animal models due to the application of MBP-11901. We plan to discontinue administration of MBP-11901 to animals in which the tumor has completely disappeared, and further investigate whether the tumor recurs through continuous follow-up.

MBP-11901, a multi-RTK inhibitor, was shown to inhibit tumor growth and kill HCC by targeting VEGFR2, c-KIT, PDGFRβ, and FLT3. Kinase profiling confirmed that there was almost no binding affinity for each of these Ras/Raf/MEK/ERK subsignals, which was further confirmed by in silico binding assays (Appendix A). Remarkably, among the various RTKs, the binding affinity to FLT3 was the strongest. Mutations in the activation loop of the tyrosine kinase domain (TKD1) of FLT3 are present in up to 30% of patients with acute myeloid leukemia (AML), and FLT3 has been utilized as a potential target for kinase inhibitor therapy [19,20]. However, FLT3 has not received much attention in HCC. A recent study showed that the reactivity of sorafenib was significantly increased in HCC with strong FLT3 expression [21]. Thus, we should focus on FLT3 as a biomarker to increase the reactivity to sorafenib, a target anticancer drug for HCC.

MBP-11901 is similar to sorafenib or cabozantinib in targeting FLT3, VEGFR2, PDGFRβ, and c-KIT. Nevertheless, it is thought that there might be additional reasons for MBP-11901 not having significant toxicity, driving tumor reduction in all mice, and leading to the complete disappearance of tumors. MBP-11901 exhibited significantly high bioavailability and long half-life pharmacokinetics in mice (data not shown). Presumably, for this reason, it has the advantage of exhibiting sufficient efficacy even at low concentrations and with no associated toxicity. We are currently conducting additional research for conducting phase 1 clinical trials.

## 5. Conclusions

In conclusion, the treatment trend for HCC has been changing. In addition to sorafenib, administration of lenvatinib has increased, as well as treatment options, including the addition of atezolizumab and bevacizumab combination therapy. However, compared with other carcinomas, HCC is difficult to cure, has limited treatment options, a poor prognosis, and a high recurrence rate, making it a highly unmet medical problem. Animal experiments confirmed the high reactivity and perfect therapeutic effect of MBP-11901, suggesting that it could be used as a new therapeutic strategy for the treatment of advanced or unresectable HCC.

## Figures and Tables

**Figure 1 cancers-14-01994-f001:**
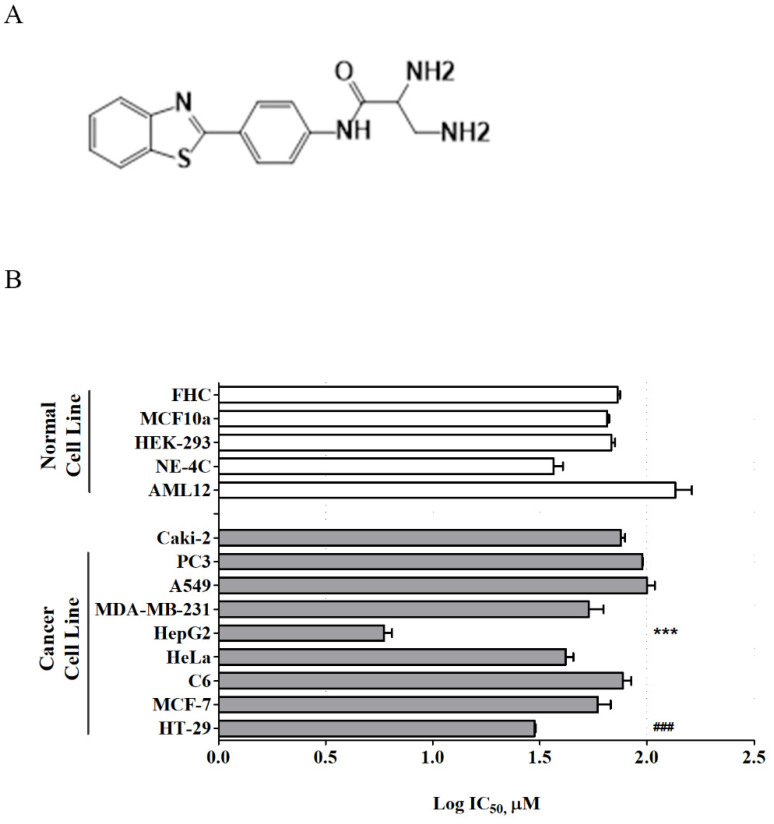
Toxicity profiles of MBP-11901. (**A**) Chemical structure of MBP-11901. (**B**) Log IC_50_ of MBP-11901 for various cancer and normal cells. Normal cells, including mouse hepatocytes (AML12), human neural stem cells (NE-4C), human embryonic kidney cells (HEK-293), human breast epithelial cells (MCF 10A), human colon epithelial cells (FHC), and a variety of human cancer cell lines, were treated with MBP-11901 (1, 5, 10, 25, 50, 75, or 100 μM) for 24 h. The log IC_50_ of 3 independent experiments per cell line was averaged and summarized as a mean. *** *p* < 0.001, significant difference from AML12. ### *p* < 0.001 significant difference from FHC.

**Figure 2 cancers-14-01994-f002:**
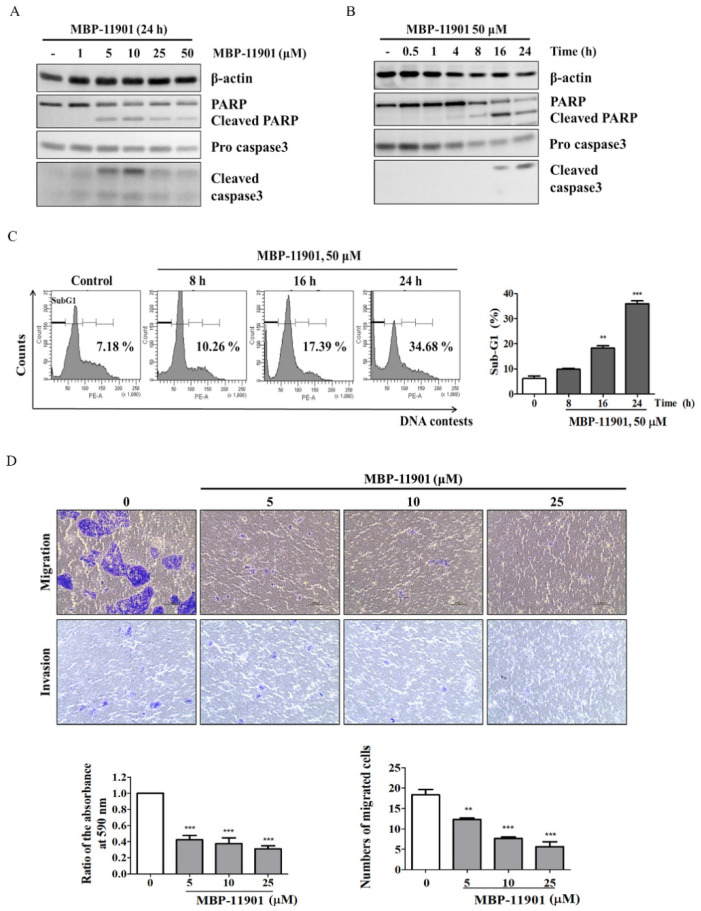
MBP-11901 induces apoptosis and inhibits cell migration and invasion of HepG2 cells. (**A**) HepG2 cells were exposed to different concentrations of MBP-11901 for 24 h and 50 μM MBP-11901 for various times (0.5, 1, 4, 8, 16, and 24 h) (**B**). Cell lysates were subjected to Western blot analysis to determine the expression of cleaved-PARP, caspase-3, and cleaved-caspase-3. β-actin was used as an internal control to monitor equal protein sample loading. The full-size blot is shown in Appendix A (**C**). HepG2 cells were treated with MBP-11901 for 8, 16, and 24 h, and analyzed for apoptotic sub-G1 by flow cytometry (**D**). The inhibitory efficacy of MBP-11901 in the migration and invasion of HepG2 cells was analyzed. The bar graph for the migration assay shows the ratio of the area of migrated cells to the total area. The bar graph for the invasion assay displays the number of cells that migrated through the Matrigel and Transwell membrane. All results were independently performed 3 times, and the average value is shown. ** *p* < 0.01, *** *p* < 0.001 relative to the control.

**Figure 3 cancers-14-01994-f003:**
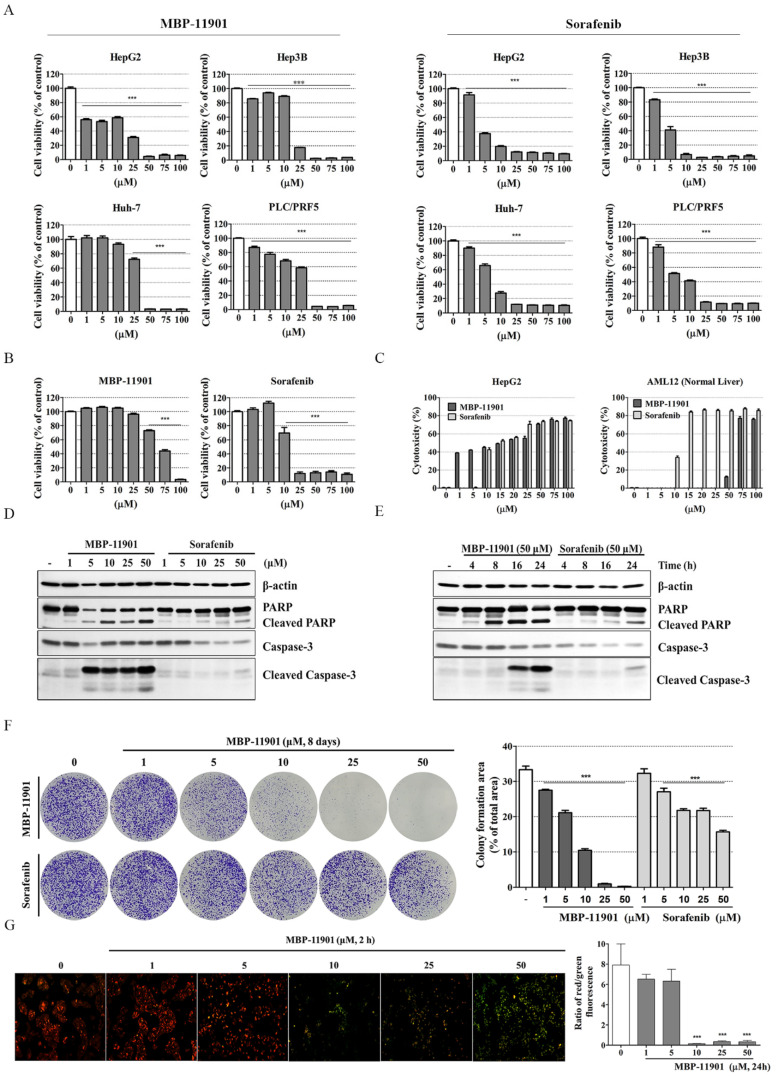
Compared with sorafenib, MBP-11901 has a positive effect on liver cancer cell lines, whereas exhibits toxicity on normal cell lines. (**A**) HepG2, Hep3B, Huh7, PLC/PRF.5, and (**B**) normal liver AML12 cells were treated with different concentrations (1, 5, 10, 25, 50, 75, or 100 μM) of MBP-11901 or sorafenib for 24 h. Cell viability was determined by the CCK-8 assay. (**C**) Cytotoxicity was analyzed by LDH leakage of HepG2 and AML12 cells. (**D**,**E**) The expression of cleaved-PARP, caspase-3, and cleaved-caspase-3 was determined in MBP-11901 or sorafenib-treated cell lysates by Western blot analysis. β-actin was used as a loading control. The full-size blot is shown in Appendix A. (**F**) Colony formation assay in HepG2 cells. The area of the colony was analyzed using the ImageJ software and plotted using GraphPad Prism. (**G**) Mitochondrial depolarization of HepG2 cells treated with MBP-11901 followed by staining with JC-1 dye was observed under a fluorescence microscope. Quantitative results of the fluorescence ratio of red to green signals. All results were independently performed 3 times, and the average value is shown. *** *p* < 0.001 relative to the control.

**Figure 4 cancers-14-01994-f004:**
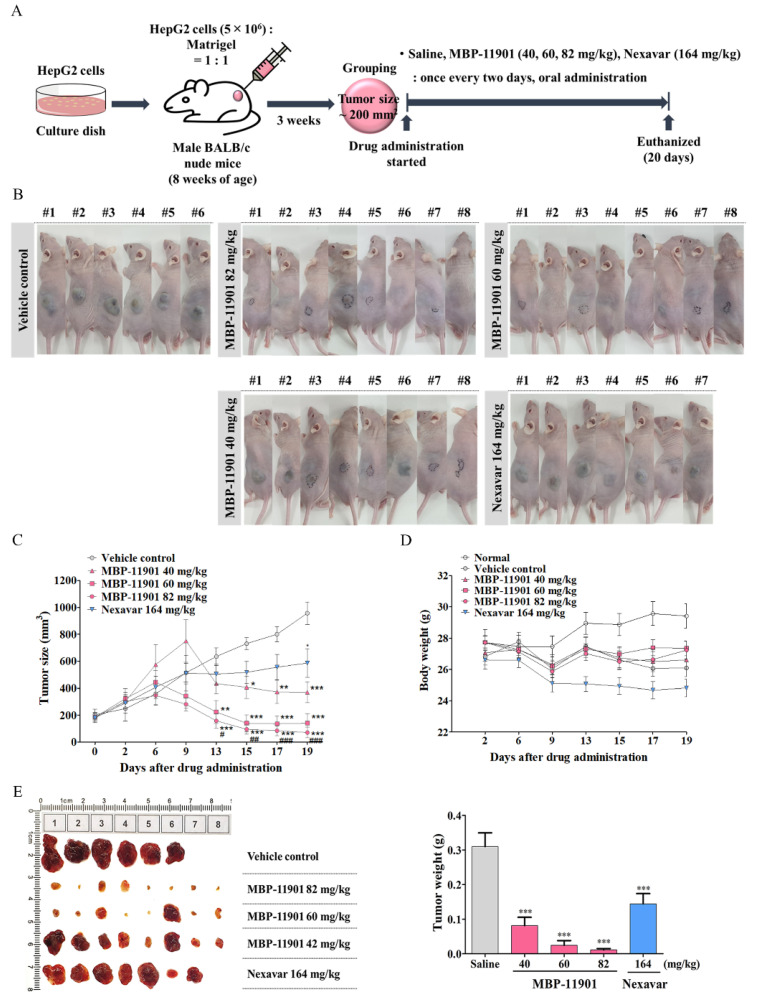
Oral administration of MBP-11901 shows an excellent anticancer effect on HCC cell-derived subcutaneous xenograft tumors in nude mice. (**A**) Experimental design: l. Mice were s.c. implanted with HepG2 cells for about 3 weeks until tumor volume reached approximately 200 mm^3^. Tumor-bearing mice were divided into 5 groups—vehicle control (saline), MBP-11901 40, 60, 82 mg/kg; and Nexavar 164 mg/kg—and administered each drug orally once every 2 d. (**B**) Images of all mice used in the experiment after a total of 9 administrations. The black line indicates the border where the xenograft tumor mass has almost disappeared. #Numbers represent individual objects. (**C**) Tumor volume and body weight were measured twice a week on fixed days. After 19 d, all mice were sacrificed and tumors were removed and weighed (**D**,**E**). The harvested tumor mass was paraffin-embedded and subjected to immunohistochemical staining. (**F**) Proliferation was assessed using a Ki-67 antibody, while (**G**) apoptosis was assessed using an antibody against cleaved caspase-3. Data shown are the mean value from mice in each group. The bar graph represents the Ki-67 or cleaved caspase-3 positive intensity for each tissue. * *p* < 0.05, ** *p* < 0.01, *** *p* < 0.001 relative to the saline group. # *p*<0.05, ## *p* < 0.01, ### *p* < 0.001 relative to the Nexavar^®^ group.

**Figure 5 cancers-14-01994-f005:**
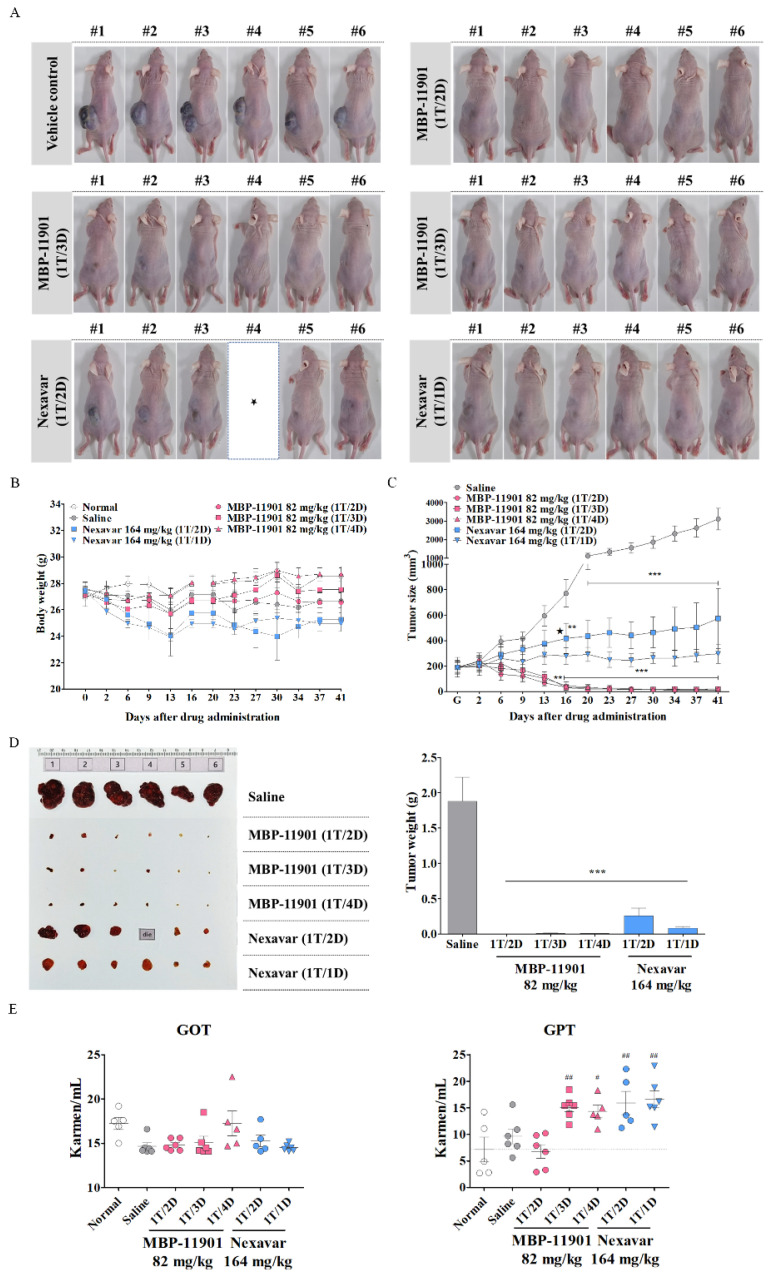
MBP-11901 shows an excellent anticancer effect even when the interval of oral administration is changed. (**A**) After 42 d, at the end of the experiment, images of subcutaneously implanted model mice were taken. # Numbers represent individual objects. (**B**,**C**) Tumor volume and body weight were measured twice a week on fixed days. (**D**) The tumors of sacrificed mice were removed, their weights were measured, and images were taken. The bar graph shows the average weight of mice in each group. Tumors completely disappeared in all subjects administered MBP-11901. (**E**) Plasma levels of the liver-specific enzymes GOT and GPT. (**F**,**G**) Representative images of H&E and Masson’s trichrome staining of livers and kidneys dissected from treated mice. Star indicates a dead animal (**A**) and the time it died (**C**). ** *p* < 0.01, *** *p* < 0.001 relative to the saline-treated control group. # *p* < 0.05, ## *p* < 0.01 relative to the normal group. 1T/1D—once a day; 1T/2D—once every 2 d; 1T/3D—once every 3 d; 1T/4D—once every 4 d.

**Figure 6 cancers-14-01994-f006:**
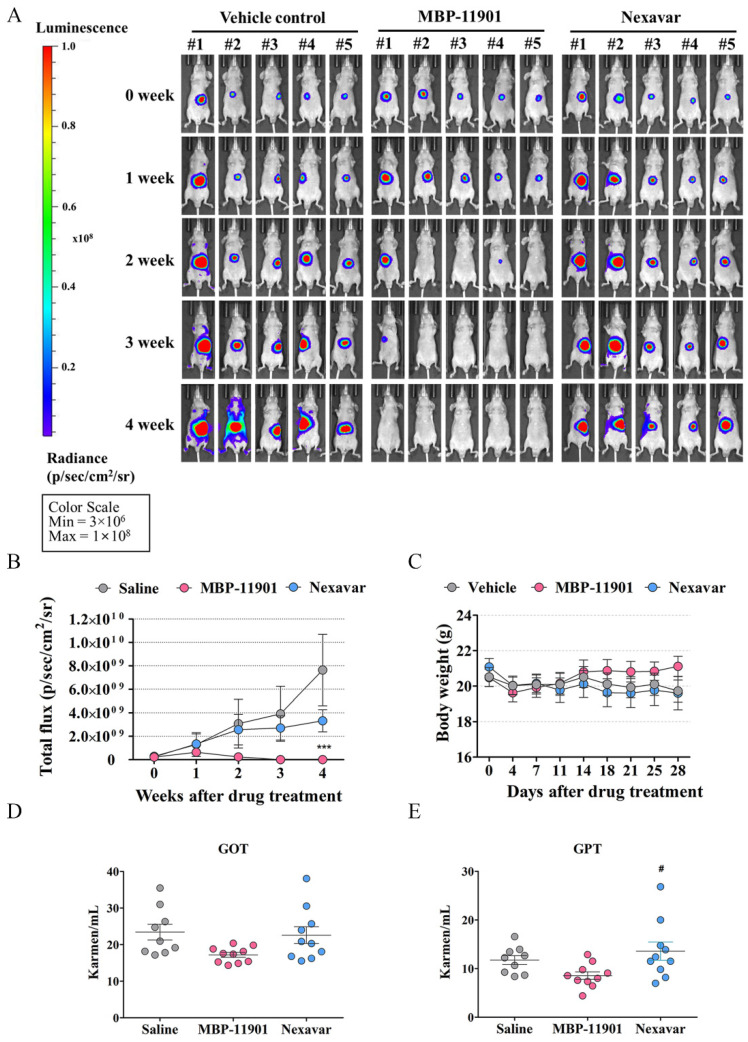
MBP-11901 induces antitumor activity in the orthotopic HCC mice model. (**A**) Bioluminescence at different times. (**B**) Quantitative analysis of IVIS signal intensity (photons/sec/cm^2^/steradian) over time after injection (*n* = 10). (**C**) Body weight. Plasma levels of the (**D**) GOT and (**E**) GPT liver-specific enzymes (*n* = 10). ***, *p* < 0.001 relative to the saline-treated control group, #, *p* < 0.05 relative to the Nexavar^®^ group. The image data for all subjects are in Appendix A.

**Figure 7 cancers-14-01994-f007:**
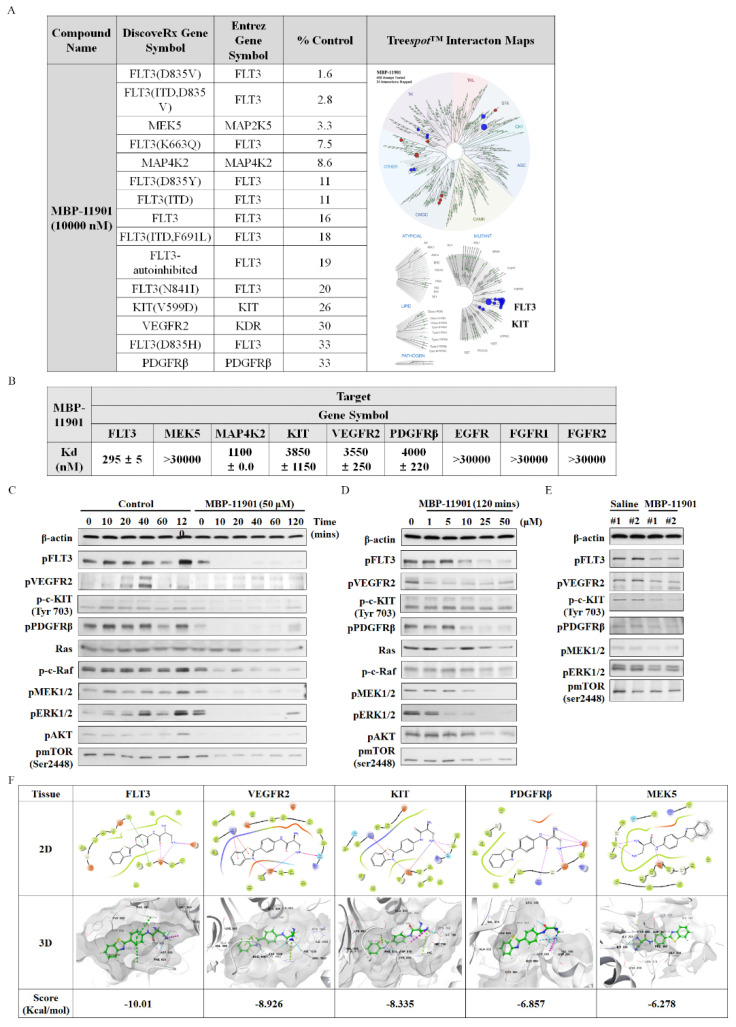
MBP-11901 induces anticancer effects on HepG2 in vitro and in vivo by inhibiting multitarget tyrosine kinases, FLT3, VEGFR2, PDGFRβ, and c-KIT. (**A**) Primary KINOMEscan profiling of 10 μM MBP-11901 using a competition binding assay for 468 human protein kinases. Kinase activities of target spectrum within Selectivity Scores (S (35%)). Results of kinase activities were reported as percent control (%Ctrl), where lower numbers indicated stronger hits. Dendrogram of human kinases showing a subset of data. (**B**) Binding interaction of MBP-11901 against selectively targeted kinases. Results of binding interaction were assessed using binding constants (Kd), where lower numbers indicated tighter binding interaction and higher affinity. Data represent the mean ± S.D. (*n* = 2). The levels of expression of FLT3, VEGFR2, PDGFRβ, c-KIT, RAS, p-c-Raf, pMEK1/2, pERK1/2, pAKT, and p-mTOR were determined by Western blotting. (**C**) Time-dependent protein expression of MBP-11901 in HepG2 cells. (**D**) Changes in proteins in HepG2 cells following timely exposure to MBP-11901. (**E**) Changes in tumor proteins after oral administration of MBP-11901 to HepG2 subcutaneously implanted xenograft mice. The full-size blot is shown in Appendix A. (**F**) 2D and 3D ligand interactions of MBP-11901 with target proteins. Hydrogen bonds—cyan dotted line; salt bridges—pink dotted line; Pi interactions—green dotted line. (**G**) A comprehensive summary of the signaling mechanisms related to the targets of MBP-11901.

**Table 1 cancers-14-01994-t001:** IC_50_ values of MBP-11901 after treatment of liver cancer cell lines for 24 h ^a^.

Tissue Sources	Cell Lines	IC_50_ (μM) for
		MBP-11901
Human hepatic carcinoma	HepG2	5.16 ± 1.37
	Hep3B	16.82 ± 0.88
	Huh-7	29.41 ± 1.13
	PLC/PRF5	18.55 ± 0.81

^a^ IC_50_ values were calculated from the percentages of cell growth inhibition obtained with 7 different concentrations (1, 5, 10, 25, 50, 75, or 100 μM) of MBP-11901.

**Table 2 cancers-14-01994-t002:** A summary of anticancer and side effects caused by MBP-11901.

	Normal	Saline	MBP-11901 40 mg/kg	MBP-11901 60 mg/kg	MBP-11901 82 mg/kg	Nexavar^®^ 162 mg/kg
Toxic deaths/total	0/5	0/6	0/8	0/8	0/8	0/7
Tumor growth inhibition rate (%)	-	0	61.41 ± 0.45 ***	85.26 ± 0.36 ***	92.54 ± 0.0 ***	38.77 ± 0.37 ***
Body weight loss rate (%)	−6.61 ± 9.41	3.84 ± 2.74	2.00 ± 2.47 ^#^	1.50 ± 1.97 ^#^	1.78 ± 4.40 ^#^	7.21 ± 3.41

Inhibition of tumor growth and rate of body weight loss were calculated based on tumor volume on day 19. *** *p* values relative to vehicle control, ^#^
*p* values relative to the Nexavar^®^-treated group.

**Table 3 cancers-14-01994-t003:** Number of doses in each group for 42 days.

Group	Total Dose Numbers
Saline	21
MBP-11901 (1T/2D)	21
MBP-11901 (1T/3D)	14
MBP-11901 (1T/4D)	11
Nexaver^®^ (1T/2D)	21
Nexavar^®^ (1T/1D)	42

**Table 4 cancers-14-01994-t004:** A summary of anticancer and side effects caused by MBP-11901.

	Normal	Saline	MBP-11901 (1T/2D)	MBP-11901 (1T/3D)	MBP-11901 (1T/4D)	Nexavar^®^ (1T/2D)	Nexavar^®^ (1T/1D)
Toxic deaths /total	0/6	0/6	0/6	0/6	0/6	1/6	0/6
Tumor growth inhibition rate (%)	-	0.00	99.82 ± 0.33 ***	99.51 ± 0.81 ***	99.48 ± 0.60 ***	85.68 ± 13.6 ***	88.02 ± 4.53 ***
Body weight loss rate (%)	−6.3 ± 6.56	10.43 ± 5.75 ^†††^	1.85 ± 3.70	−0.43 ± 3.76 ^#^	−4.29 ± 0.59 ^##^	10.11 ± 8.45 ^†††^	9.30 ± 4.43 ^†††^

Inhibition of tumor growth and rate of body weight loss were calculated based on tumor volume on day 41. ^†††^
*p* values relative to the normal group, *** *p* values relative to the saline-treated control group, and ^#^, ^##^
*p* values relative to the Nexavar^®^ (1T/2D, 1T/1D)-treated group.

## Data Availability

The data presented in this study are available in this article.

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
