# Peer review of "MBP-11901 Inhibits Tumor Growth of Hepatocellular Carcinoma through Multitargeted Inhibition of Receptor Tyrosine Kinases"

_cancers, 2022, doi:10.3390/cancers14081994_

Round 1

Reviewer 1 Report

In this manuscript, the authors have investigated the anti-tumor effect of MBP-11901, a new synthesized benzothiazole aniline derivatives both in vitro and in vivo. They investigated the toxicity of MBP-11901 among various normal and cancer cell lines and found MBP-11901 showed the most potent cytotoxicity to human HCC and colon cancer cells compared with corresponding normal cells. By utilizing multiple HCC cell lines and different mouse models, their in vivo results showed strong inhibition of tumor growth and even tumor disappearance, indicating its promising anti-tumor effect.  Lastly, they discovered that MBP-11901 inhibited tumor growth though inhibition of multiple tyrosine kinases, including FLT3, VEGFR2, PDGFR beta. Overall, their results suggested that MBP-11901 could serve as a new therapeutic strategy for the treatment of HCC. However, there are some remaining questions to be answered:

1, In Fig 7, the authors observed decreased phosphorylation of several kinases by western blotting. Have the authors examined the total protein level of these kinases? Does MBP-11901 also affect the expression of these proteins?

2, It is interesting that GPT was elevated with MBP-11901 treatment every 3 or 4 days, not every 2 days in vivo. Increased GPT level in plasma indicated damaged liver, however, this phenomenon is not observed when drug treatment was given in short interval. Could the authors discuss why 2 days interval of MBP-11901 treatment has no impact on GPT level while long interval treatment (3 or 4 days) instead increased GPT level dramatically?

3, In Fig 3A, 5uM Sorafenib treatment showed only ~40% cell viability on HepG2, however, Fig 3C indicated that 5uM Sorafenib didn’t exhibit any cytotoxicity on HepG2.

4, Line 296, mm3 not mm3

Author Response

Hello reviewer
All modified files cannot be transferred online due to their large size. The added file has been sent to the CANCERS editor.
please check.

The answers to the questions are as follows.

Revision to CANCERS

We thank the referees for their very thoughtful reviews.

We are also deeply grateful to the editors of CANCERS who have been active in helping our paper to be reviewed quickly.

We have carefully considered the comments and have revised the manuscript (cancers-1659720) accordingly. Our specific responses are detailed below. You should refer to the file title [cancer-1659720-manuscript with track].

Respectfully,

Response to reviewer #1

Q1) In Fig 7, the authors observed decreased phosphorylation of several kinases by western blotting. Have the authors examined the total protein level of these kinases? Does MBP-11901 also affect the expression of these proteins?

: Thank you for your interest. It is natural that you are curious about changes in total protein. It is regrettable that we would have liked to have presented the total protein change of all phosphorylated proteins together in this submission. We did our best to perform additional Western blots on the identifiable portion of the total protein. We are sorry that the remaining lysate is not sufficient and it is difficult to obtain antibodies.

Added Western blot data. No pictures are added. Please check the file passed to the editor.

As can be seen from the obtained band results, in the MBP-11901 concentration-dependent result, there was no significant change in most of the total proteins performed except for slight changes in the total proteins of FLT3 and c-KIT. It can be seen that the change in phosphorylated protein is very clear compared to the change in Total protein.

Q2) It is interesting that GPT was elevated with MBP-11901 treatment every 3 or 4 days, not every 2 days in vivo. Increased GPT level in plasma indicated damaged liver, however, this phenomenon is not observed when drug treatment was given in short interval. Could the authors discuss why 2 days interval of MBP-11901 treatment has no impact on GPT level while long interval treatment (3 or 4 days) instead increased GPT level dramatically?

: We are very grateful for your comment. The part you suggested is the part that we are also paying attention to.

There is no convincing evidence to suggest the cause of the rise in GPT when administered at intervals of 3 or 4 days rather than at intervals of 2 days. However, whether this phenomenon is temporary, variation of the mouse object used, the environment in which the experiment was performed, or whether it is a pharmacokinetic result must be confirmed through additional repeated tests.

Currently, pharmacokinetic data from non-clinical rats are in the final stages of collection. In addition, we are planning additional tests using conditions that change the concentration and dosing interval at the same time.

If the hepatotoxicity problem due to GPT elevation is in mind, we can confirm that MBP-11901 has a wide margin of safety in the results of single and repeated toxicity studies (GLP) that we recently completed.

Test Items

Progress

Summary of results

Single

dose toxicity

SD rat

complete

Ÿ [lethal dose] 1000-2000 mg/kg

Beagle

complete

Ÿ [Maximum tolerated dose] 40 mg/kg

Repeated

dose toxicity

[Rat] 2-week Dose Range Finding (DRF)

complete

Ÿ 500, 250, 125 mg/kg

[Rat] 4-week Repeated Dose Toxicity and 2-week Recovery Study

complete

Ÿ [No Observed Adverse Effect Leve] 125 mg/kg

[Rat] 4-week Repeated Dose Toxicity and Toxicokinetic Study

complete

Ÿ AUC, Cmax, Tmax analyzed. No accumulation by repetition

[Beagle] 2-week Dose Range Finding (DRF)

complete

Ÿ 100, 30, 10 mg/kg

[Beagle] 4-week Repeated Dose Toxicity and 2-week Recovery Study

Post-mortem

Ÿ At high doses, there are no specifics other than intermittent vomiting.

[Beagle] 4-week Repeated Dose Toxicity and Toxicokinetic Study

Post-mortem

Q3) In Fig 3A, 5uM Sorafenib treatment showed only ~40% cell viability on HepG2, however, Fig 3C indicated that 5uM Sorafenib didn’t exhibit any cytotoxicity on HepG2.

: I am very grateful for your advice. We tried to confirm the cytotoxicity in cancer cell lines and normal cell lines by various methods. Figure 3A is a method of measuring cell viability according to mitochondrial dehydrogenase enzyme activity in living cells. Figure 3C is an experiment to confirm that lactate dehydrogenase (LDH) is leaked out of the cell due to cell membrane changes due to cytotoxicity and is cytotoxic. Both methods are commonly used to measure cytotoxicity, but because the target enzyme is different, the resulting cytotoxicity level may be different. Although CCK-8 is not cytotoxic, it is sometimes toxic to LDH assays and vice versa. We tried to check how sensitive MBP-11901 and sorafenib are to cancer cells and low toxicity to normal cells compared to control conditions using the above two methods.

Q4) Line 296, mm3 not mm3

: I am very grateful for your notion of the mistake I made. Line 296 has been changed to line 301 due to the previous correction, and mm3 of line 301 has been changed to mm3. Thank you.

Reviewer 2 Report

The manuscript entitled "MBP-11901 Inhibits Tumor Growth of Hepatocellular Carcinoma through Multitargeted Inhibition of Receptor Tyrosine 3 Kinases" introduces a newly synthesized Benzothiazole Aniline Derivative with promising anticancer activity against human hepatocellular carcinoma (HCC). They demonstrated that MBP-11901 inhibited cell proliferation, migration and colony-formation capacity of human HCC-derived cell lines, but even do not have negative impact on healthy murine hepatocytes. In addition, oral administration of MBP-11901 significantly blocked tumor growth in different xenographted-HCC mouse models in comparison with control and sorafenib-treated groups. Methods are well-described and appropiate, and the manuscript is easy to read. Therefore, I just have minor concerns to accept this paper:

  • Number of animals in the orthotopic-model of HCC is missing
  • I would suggest to increase the quality of the figures. Some of them contain very small letters that are hard to see.
  • If possible, repeat the cytotoxicity/cell viability assays in healthy human hepatocytes or healthy hepatic cell line (for example, HepaRG or LO2)
  • Correct "caboxantinib"; change it to "caboZantinib".

Author Response

Hello reviewer
All modified files cannot be transferred online due to their large size. The added file has been sent to the CANCERS editor.
please check.

The answers to the questions are as follows.

Revision to CANCERS

We thank the referees for their very thoughtful reviews.

We are also deeply grateful to the editors of CANCERS who have been active in helping our paper to be reviewed quickly.

We have carefully considered the comments and have revised the manuscript (cancers-1659720) accordingly. Our specific responses are detailed below. You should refer to the file title [cancer-1659720-manuscript with track].

Respectfully,

Response to reviewer #2

Q1) Number of animals in the orthotopic-model of HCC is missing

: Originally, we performed experiments on 10 individuals in each group. During the experiment, due to a handling error, the group #7 of Saile was lost, so Salie acquired data for a total of 9 animals. Although the Manuscript shows images for 5 animals, fluorescence intensity and GOT/GPT analysis are results for a total of 10 animals. Image data for all subjects were added to supplementary Figure 8.

There are additional animal test fluorescence image datas. No pictures are added. Please check the file passed to the editor.

Q2) I would suggest to increase the quality of the figures. Some of them contain very small letters that are hard to see.

: I have edited all the pictures in the text (except for supplementary figures). The font has been changed to Times New Roman, and the font has been increased by 2~4 fonts or more. Even with this effort, it is not so obvious that the text grows larger as many pictures fit on one page.

We asked the editor to help exclude the phenomenon of small text or pictures by dividing many pictures into two pages.

Q3) If possible, repeat the cytotoxicity/cell viability assays in healthy human hepatocytes or healthy hepatic cell line (for example, HepaRG or LO2)

: Unfortunately, we do not have a healty human hepatocyte cell line, nor do we have a quick way to purchase it, so we were unable to conduct further experiments. I'm very sorry.

If you are also concerned about the toxicity to normal cells, please refer to our non-clinical GLP toxicity test results.

Test Items

Progress

Summary of results

Single

dose toxicity

SD rat

complete

Ÿ [lethal dose] 1000-2000 mg/kg

Beagle

complete

Ÿ [Maximum tolerated dose] 40 mg/kg

Repeated

dose toxicity

[Rat] 2-week Dose Range Finding (DRF)

complete

Ÿ 500, 250, 125 mg/kg

[Rat] 4-week Repeated Dose Toxicity and 2-week Recovery Study

complete

Ÿ [No Observed Adverse Effect Leve] 125 mg/kg

[Rat] 4-week Repeated Dose Toxicity and Toxicokinetic Study

complete

Ÿ AUC, Cmax, Tmax analyzed. No accumulation by repetition

[Beagle] 2-week Dose Range Finding (DRF)

complete

Ÿ 100, 30, 10 mg/kg

[Beagle] 4-week Repeated Dose Toxicity and 2-week Recovery Study

Post-mortem

Ÿ At high doses, there are no specifics other than intermittent vomiting.

[Beagle] 4-week Repeated Dose Toxicity and Toxicokinetic Study

Post-mortem

Q4) Correct "caboxantinib"; change it to "caboZantinib".

: As mentioned above, every “caboxantinib” has been replaced with every "caboZantinib" (in line 605, 687).
